# Physical Activity, Mental Health, and Quality of Life among School Students in the Jazan Region of Saudi Arabia: A Cross-Sectional Survey When Returning to School after the COVID-19 Pandemic

**DOI:** 10.3390/healthcare11070974

**Published:** 2023-03-29

**Authors:** Mohamed Salih Mahfouz, Ahmad Y. Alqassim, Nasser Hadi Sobaikhi, Abdulaziz Salman Jathmi, Fahad Omar Alsadi, Abdullah Mohammed Alqahtani, Mohammed Mohalhil Shajri, Ibrahim Darwish Sabi, Ahmed M. Wafi, Jonathan Sinclair

**Affiliations:** 1Family and Community Medicine Department, Faculty of Medicine, Jazan University, Jazan 45142, Saudi Arabia; 2Faculty of Medicine, Jazan University, Jazan 45142, Saudi Arabia; 3Physiology Department, Faculty of Medicine, Jazan University, Jazan 45142, Saudi Arabia; 4Research Centre for Applied Sport, Physical Activity and Performance, Faculty of Allied Health and Wellbeing, School of Sport & Health Sciences, University of Central Lancashire, Preston PR1 2HE, UK

**Keywords:** depression, pediatric quality of Life, environmental health, sport and children

## Abstract

Increasing evidence suggests that physical activity (PA) can reduce depression and anxiety in adolescents. At the same time, quality of life (QoL) is sensitive to both mental health and PA, but little is known about the mechanism between these three variables among adolescents. This study aimed to assess the physical activity, mental health, and quality of life of school students when they returned to school after two years of distance learning in the Jazan region. This current investigation represented an observational cross-sectional survey conducted in January 2022 among a random sample of 601 students from intermediate and high schools in the Jazan region, Saudi Arabia. Three standardized questionnaires were used for data collection; the Arabic version of the Pediatric Quality of Life Inventory (PedsQL), Depression Anxiety Stress Scales (DASS21), and the Fels PAQ for children. The analysis revealed a moderate level of physical activity, decreased HRQoL, and symptoms of mental health problems (anxiety, depression, and stress) among the schools’ students when they returned to school following COVID-19 lockdown. The overall Pediatric Quality of Life mean score was (81.4 ± 16.4), which differed significantly according to gender, age groups, and grade levels (*p* < 0.05 for all). There was a negative correlation between the overall quality of life and mental health domains. Sport was negatively correlated with mental illness symptoms and positively correlated (*p* < 0.05) with Pediatric Quality of Life. The regression models revealed that stress was a significant predictor for the quality of life of male and female adolescents ([β = −0.30, (95% CI (−0.59) to (−0.02), *p* < 0.05)] and [β = −0.40, (95% CI (−0.70) to (−0.01), *p* < 0.05)], respectively). The analysis revealed a moderate level of physical activity among the schools’ students when they returned to school following COVID-19 lockdown. Children’s involvement in physical activity was associated with improved quality of life and mental health. The results call for the need to develop appropriate intervention programs to increase school students’ physical activity levels.

## 1. Introduction

The COVID-19 pandemic has significantly altered people’s daily lives, compelling countries to implement school closures and replacements with virtual classes, lockdowns, isolation, and social distancing to contain the virus’s spread and mitigate its adverse health consequences [1,2]. Additionally, COVID-19 has greatly impacted daily life, influencing how individuals live, work, learn, interact, engage in physical activity, and manage their mental health

Physical activity refers to any body movement generated by skeletal muscles that require energy expenditure [3]. How to encourage children and adolescents to perform an appropriate level of PA has always been a major public health concern that was worsened by the pandemic [4]. According to the recommendation given by the WHO, to achieve health benefits, children and adolescents should do 60 min or more of moderate-to-vigorous-intensity PA each day [5,6].

Recent research [7,8] suggests that the COVID-19 pandemic may exacerbate physical inactivity and prolong sedentary time in children, which may be attributed to the lack of equal opportunities to participate in PA and the unavailability of sports facilities caused by school closures. It has been shown that students have lower levels of PA and more sedentary time during the weekends than in school [9].

The COVID-19 pandemic also poses a significant mental health threat among children, adolescents, and all students generally [10]. It has caused tremendous stress levels, primarily due to schools being closed and home quarantining. As a result, children have become physically less active, have much-prolonged screen times, have irregular sleep schedules, and have less healthy diets, resulting in excess weight and a lack of cardiorespiratory performance [10]. Students exposed to these incidents can trigger the development of anxiety, panic attacks, depression, and mood disorders, and it may also lead to a deterioration in mental health [11].

A prior study of parents in Italy and Spain discovered that more than 85% of their children decreased their physical activity, increased their screen times, and increased maladaptive emotional and behavioral indicators [11]. According to the Centers for Disease Control and Prevention, 1.9 million children between the ages of 3 and 17 years have been diagnosed with depression, and 4.4 million have been diagnosed with anxiety as a result of home quarantine in the USA [12]. Moreover, mental illness and physical inactivity among students are not just limited to COVID-19 lockdown but extend to the post-lockdown period as well. Recent research from China suggests that students and adolescents reported significantly more mental health problems after the COVID-19 school lockdowns [13].

Many studies have documented that PA is positively associated with mental health, while an inverse association has been reported between psychiatric distress (depression, stress, and anxiety) and quality of life among adolescents [14,15]. Additionally, physical inactivity has been found to be inversely associated with quality of life. Despite the inter-variable associations between these three important variables, the mechanisms underlying these associations have not been investigated after the COVID-19 pandemic lockdown. Therefore, the objectives of this study are: (i) to assess the physical activity, mental health, and quality of life of school students when they returned to school after two years of distance learning in the Jazan region, southwest Saudi Arabia; (ii) to investigate the association between physical activity, mental health, and quality of life among the students; and (iii) to determine whether any differences exist between students in terms of physical activities, mental health, and quality of life according to gender. The study results will shed light on adolescents’ quality of life, physical activity, and mental health post COVID-19 lockdown and offer policymakers and school administrations guidance regarding these essential indicators.

## 2. Materials and Methods

### 2.1. Study Design, Setting, and Population

A cross-sectional survey was conducted among students of intermediate and high schools in the Jazan region. School-aged children and adolescents (intermediate and secondary schools) are an important segment of the population, and additional information was needed to determine their mental health, QoL, and PA statuses post COVID-19 period. The Jazan region is one of the KSA’s thirteen regions. It is located on the tropical Red Sea Coast in southwestern KSA. The region covers an area of 11,671 km^2^, including some 5000 villages and towns, with a population of 1.5 million. The intermediate educational stage in Saudi Arabia lasts three years, while secondary education lasts for the same period. During the COVID-19 pandemic, around 22,000 schools were closed from 9 March 2020. Classrooms opened again for students in intermediate and high schools on 29 April 2021. Inclusion criteria involved school-age children (intermediate and secondary schools) who registered for the academic year 2021/2022, lived in the Jazan region, and were aged between 12 to 18 years old. This research was conducted in January 2022.

### 2.2. Sampling Procedures

A sample size of 640 students was determined for conducting this survey. The sample size estimation was based on Cochran’s formula [16] for cross-sectional surveys: initial sample size n = [(z^2^ * p * q)]/d^2^, where p is the (estimated) proportion of the population, which has the attribute in question; q = (1 − p); d is the desired level of precision; and Z is the critical value of the normal distribution at the required confidence level; usually, Z = 1.96 for 95% C.I. This research utilized the following indicators: p = prevalence of the required, studied phenomenon, which was set at 50% (as no information related to the main research outcome was available); Z = 1.96 confidence interval; and d = error not more than 4%, with a 10% non-response rate. Due to the pandemic situations of COVID-19, the research combined both the random and non-random sampling designs to recruit the students for this study. In the first stage, eight schools (two intermediate and two high schools for both genders) were selected randomly from each educational sector. The Jazan region, administratively, is divided into two educational sectors. Second, a convenience sample was utilized to select the pupils from each selected school to participate in this study. The survey link was sent to the child’s parent/guardian, and the parents/guardians were asked to assent to their child’s participation in the survey before the child could participate.

### 2.3. Data Collection and Instrumentation

The study was conducted using a self-administered questionnaire prepared in Arabic and distributed via an anonymous online survey instrument. The first part of the questionnaire covered the socio-demographic information of the study participants, such as age, gender, nationality, residence, and parent’s education level. The second part of the questionnaire involved the validated Arabic version of the Fels PAQ for children (7–19 years). The instrument was a self-administered questionnaire assessing habitual physical activity in an eight-item questionnaire containing three “open” questions for which some activities required listing by the study participants. The frequency of participation for each activity was also obtained [17]. The third part of the questionnaire was the Arabic version of the Depression Anxiety Stress Scales (DASS21) [18]. DASS is a 21-item instrument measuring current (“over the past week”) depression, anxiety, and stress symptoms. Participants were asked to use a 4-point combined severity/frequency scale to rate the extent to which they experienced each item over the past week. The scale ranged from 0 (did not apply to me at all) to 3 (applied to me very much or most of the time). Scores for depression, anxiety, and stress were calculated by summing the relevant item scores of depression, anxiety, and stress. The fourth part of the questionnaire was the Arabic version of the Pediatric Quality of Life Inventory (PedsQL) [19], which is a 23-item generic health status instrument with child forms that assesses five domains of health (physical functioning, emotional functioning, psychosocial functioning, social functioning, and school functioning) in children and adolescents aged 13 to 18. A pilot study was conducted among 30 participants from one school to test the questionnaire’s applicability and understanding before starting the actual research. The assessment of the internal consistency for the three measures showed Cronbach’s α = 0. 932 for the PedsQL; DASS21 showed a Cronbach’s α of 0.954; and finally, the FELS PAQ for children recorded a Cronbach’s α = 0.867.

### 2.4. Study Ethics

The research was conducted according to the ethics guidelines of Saudi Arabia. Ethical clearance was obtained from the Standing Committee for Scientific Research Ethics-Jazan University (HAPO-10-Z-001) (REF# REC-43/05/088). All study participants read and signed the consent form following a short introduction about the study objectives. Participants were told that they had the right to withdraw at any time and that there would not be any harm or loss of benefits if they continued or withdrew from participating. Finally, we strictly ensured that each participant’s personal information was preserved and confidentiality was maintained.

### 2.5. Statistical Analysis

The IBM SPSS Statistics for Windows, Version 24.0. Armonk, NY: IBM Corp program was used for data analysis. The analysis involved descriptive as well as inferential statistics according to the essential purpose of each relationship. Categorical variables were described as frequencies and percentages. The normality of the continuous variables was assessed using the Kolmogorov–Smirnov test. To assess the differences in the demographic characteristics in the FELS PAQ and PedsQL, we compared the means of continuous variables using the Student’s t-test and one-way ANOVA, respectively. The effect size, based on Cohen’s d and omega squared, was calculated. Cohen’s d was interpreted as follows: <20 as very small; 0.20–0.49 as small; 0.51–0.80 as intermediate; and >0.8 as a large effect. An omega-squared effect size of 0.01–0.06 was considered small, 0.06–0.14 was considered medium, and >0.14 was considered large. A multiple linear regression model was used to assess predictors of the overall score of quality of life among the pupils. The final model was assessed for multicollinearity and assumptions of the ordinary least squares (OLS) technique. The Pearson correlation coefficient was computed to assess the correlation between the overall quality of life, sport index, and the DASS21 domains among the adolescents. The correlation coefficient was defined as weak (0–0.29), moderate (0.30–0.50), and strong (>0.50). A *p*-value less than 0.05 was considered significant.

## 3. Results

Table 1 summarizes the study participants’ characteristics and the levels of the HRQoL and FELS PAQ scores. Among the pupils, males constituted 337 (56.1%). Most of them were in the age group of 17–18 years, and 460 (76.5%) lived in rural areas. Regarding their parents’ education levels, 273 (53.8%) of the fathers were of secondary and university education, compared to 218 (42.9%) of the mothers. Almost 377 (74.2%) of the mothers were housewives, while more than one-third of the fathers, 196 (38.6%), were working in governmental positions. The table further showed that the PedsQoL scores differed significantly according to gender, age groups, and grade levels, whereas the FELS PAQ total score was significantly different according to grades only (*p* < 0.05 for all).

Table 2 presents descriptive statistics for mental health, physical activity, and quality of life for boys and girls. According to the table, girls scored significantly higher than boys on all three DASS subscales. Boys had a significantly higher sports index than girls (*p* < 0.001). According to the Pediatric Quality of Life (PedsQL), the table showed that boys had significantly higher scores in all PedsQL domains (*p* < 0.05), with a small effect size ranging between 0.169–0.450.

Regarding mental health among the students, 341 (56.7%) were assessed to be free of depression symptoms; 67 (11.1%) suffered from mild depression; 99 (16.5%) had moderate depression; and only 45 (7.5%) and 49 (8.2%) were considered to have severe and extremely severe depression symptoms, respectively. Regarding the anxiety subscale, 41 (51.6%) were normal; 41 (6.8%) suffered from mild anxiety; 105 (17.5%) had moderate anxiety; and 49 (8.2%) and 96 (16.0%) were considered to suffer from severe and extremely severe anxiety, respectively. Finally, for the self-reported stress majority, 415 (69.1%) were normal; 63 (10.5%) were considered to have mild stress; 49 (8.2%) were classified as having moderate stress symptoms; and only 43 (7.2%) and 31 (5.2%) were considered to suffer from severe and extremely severe stress, respectively. The PedsQL scores were found to be significantly different across all DASS21 self-reported subscales (*p* < 0.05) [Table 3].

Table 4 shows the correlations between adolescents’ quality of life, sports, and mental health (DASS21 domains). According to the table, there was a significant, strong negative correlation between the overall quality of life and all mental health domains (*p* < 0.05). Sports showed weak negative correlations with all mental health domains (*p* > 0.05) and a weak positive correlation with the overall quality of life (*p* > 0.05).

Table 5 shows the results of the multiple linear regression models, which demonstrated that stress was a significant predictor for the low quality of life of male and female adolescents ([β = −0.30, (95% CI (−0.59) to (−0.02), *p* < 0.05)] and [β = −0.40, (95% CI (−0.70) to (−0.01), *p* < 0.05), respectively]).

Figure 1 shows the sports played in school by the adolescents. It was found that the percentage of those who never practiced sports was 78.7% for tennis, 74.4% for basketball, 69.2% for volleyball, and 51.3% for football. The percentage who practiced football regularly was 25.0%, compared to 13.3% for volleyball, 14.3% for basketball, and 12.5% for tennis.

## 4. Discussion

The present study aimed to assess the mental health, physical activity, and quality of life of school students when they returned to school after two years of distance learning in the Jazan region, southwest Saudi Arabia. Children and adolescents were at higher risk of mental problems due to restrictions during the pandemic. For that, families were greatly challenged to adapt to these restrictions and to create a healthy environment, particularly during online classes.

During the pandemic, the government established quarantine measures to control the spread of the disease; this led to restrictions on freedom and significantly increased stress among the population. These restrictions also led to violence, in some instances, among families and may have had an impact on the mental health of children and adolescents [20,21], eventually leading to long-term adverse outcomes. Our findings suggested that the COVID-19-related lockdown had a negative effect on children’s and adolescents’ mental health and physical activity. Additionally, the results concluded that this period had a negative impact on the pupils’ quality of life.

Regular physical activity and exercise are related to an improved sense of well-being, physical health [22], life satisfaction, and cognitive performance. Our results revealed a high proportion of pupils who never practiced any type of sport. This pattern of low level of PA was not surprising, as the COVID-19 period affected the involvements of the students in all types of sports. This pattern of PA behavior was entirely consistent with expectations and the adolescents still affected by the COVID-19 period, which is characterized by a low level of physical activity [23]. In contrast to a previous study [23], we found significant gender differences in physical activity, with girls being less active than boys, but the reasons for this remain unclear. Previously, these gender differences were explained by non-modifiable elements, such as a girl’s biology, as well as by certain modifiable variables, such as a girl’s psychology, social support, and cultural and physical environmental factors. As is the case in most Arab countries, girls receive less social support than boys, as many families limit their girls from exercising outdoors, due to traditional cultural and religious norms [6]. As a result, authorities, schools, health, exercise providers, and families must be aware of the severity of the problem and quickly establish appropriate physical activity interventions to mitigate the detrimental effects of the COVID-19 outbreak on children’s and adolescents’ health [24].

A significant finding of our study was that depression, stress, and anxiety levels all had negative effects on the overall quality of life and physical activity; these findings were consistent with a study in Italy and Spain that reported that 85.7% observed a change in the behaviors and emotions of their children during the pandemic, based on parents’ reports [11]. Another cross-sectional study was conducted in China, which targeted adolescents during the pandemic, and they assessed depression using the Patient Health Questionnaire (PHQ-9) and anxiety using Generalized Anxiety Disorder (GAD-7), both of which were not used in our study. The prevalence of depression and anxiety were 34.7% and 37.4%, respectively [25].

The results suggested that 43.3% of participants were depressed, and overall quality of life correlated with depression. These outcomes were also found in a study conducted in China among adolescents, which reported that 39.5% of the participants experienced depression, which was linked with less physical activity as a factor of depression during the COVID-19 pandemic [26].

In this study, quality of life involved the physiological integrity of individuals, as well as mental health and social comfort. The analysis revealed that parameters of quality of life were affected negatively by psychological issues. Additionally, males and females with higher physical activity levels had considerably higher quality of life scores. Regular physical activity was established in literature to considerably improve the quality of life of healthy individuals and chronic disease patients. In line with this approach, the results of our study’s multiple linear regression analysis revealed that students’ mental health had a negative effect on their quality of life, particularly depression, which is a major predictor of both male and female teenage quality of life. These findings highlighted the critical significance of including physical exercise and mental health programs in community-based rehabilitation programs to safeguard and maintain patients’ quality of life during the ongoing pandemic [23]. This study highlighted the adverse impact of COVID-19 on physical activity and the mental health of adolescents. The results can help inform decision-makers about the relationship between the PA, HRQol, and mental health and pave the way for the formulation of intervention programs that can improve adolescents’ quality of life.

The present study had some limitations. This was a cross-sectional-based study. Therefore, the temporal correlation between explanatory variables and outcomes could not be assessed. As this was an online survey-based study, response and selection biases are possible. Finally, the results of our study cannot be generalized to the whole population in the Jazan Region.

## 5. Conclusions

The analysis revealed a moderate level of physical activity among the schools’ students when they returned to school following COVID-19 lockdown. Children’s physical activity involvement is linked to the improvement in quality of life and mental health. The results call for the need to develop appropriate intervention programs to increase school students’ physical activity levels. Future research focusing on the inter-variable associations between PA, mental health, and quality of life, using higher study designs, such as a cohort study, is highly needed.

## Figures and Tables

**Figure 1 healthcare-11-00974-f001:**
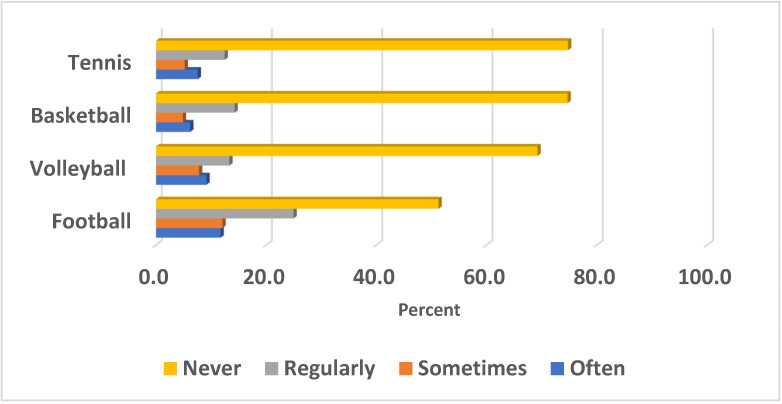
Sports played in school by adolescents.

**Table 1 healthcare-11-00974-t001:** Background characteristic, health-related quality of life, and physical activity scores among the study participants (n = 601).

Characteristics		N	%	PedsQoL Scores		FELS PAQ Total Score	
				Mean ± SD	*p*-Value	Mean ± SD	*p*-Value
Gender	Male	337	(56.1)	84.1 ± 16.0	<0.001	9.5 ± 1.8	0.203
	Female	264	(43.9)	77.8 ± 17.5		9.4 ± 1.6	
Age groups	12–14 year	86	(14.3)	76.7 ± 17.8	0.010	9.6 ± 1.6	0.304
	15–16 years	189	(31.4)	80.9 ± 16.2		9.6 ± 1.7	
	17–18 years	326	(54.2)	82.9 ± 17.0		9.4 ± 1.8	
Residence	Village	460	(76.5)	80.8 ± 17.1	0.148	9.4 ± 1.7	0.279
	Town	141	(23.5)	83.2 ± 16.3		9.6 ± 1.9	
Grades	1st Intermediate	43	(7.2)	72.0 ± 16.6	<0.001	9.9 ± 1.6	0.300
	2nd Intermediate	43	(7.2)	79.7 ± 21.5		9.5 ± 1.5	
	3rd Intermediate	52	(8.7)	77.3 ± 16.2		9.6 ± 1.9	
	1st High School	164	(27.3)	82.5 ± 15.2		9.6 ± 1.8	
	2nd High School	158	(26.3)	85.0 ± 15.0		9.1 ± 1.6	
	3rd High School	141	(23.5)	80.9 ± 18.4		9.5 ± 1.9	
Father’s Level of Education	Primary	118	(23.2)	80.0 ± 16.3	0.508	9.4 ± 1.7	0.079
	Intermediate	68	(13.4)	80.2 ± 17.4		10.1 ± 1.7	
	Secondary	140	(27.6)	80.9 ± 17.9		9.4 ± 1.7	
	University	133	(26.2)	82.5 ± 15.2		9.5 ± 1.6	
	Postgraduate	49	(9.6)	77.8 ± 17.7		9.5 ± 1.9	
Mother’s Level of Education	Primary	184	(36.2)	80.7 ± 16.5	0.922	9.5 ± 1.7	0.850
	Intermediate	72	(14.2)	80.3 ± 16.9		9.5 ± 1.7	
	Secondary	112	(22.0)	80.7 ± 18.5		9.5 ± 1.8	
	University	106	(20.9)	81.7 ± 14.2		9.6 ± 1.6	
	Postgraduate	34	(6.7)	78.7 ± 19.1		9.2 ± 1.9	
Father’s Job	Own Business	45	(8.9)	81.7 ± 16.4	0.173	9.7 ± 1.8	0.751
	Government Sector	196	(38.6)	80.5 ± 15.8		9.5 ± 1.6	
	Private Sector	26	(5.1)	73.7 ± 14.6		9.9 ± 1.6	
	Retired	167	(32.9)	80.7 ± 18.9		9.5 ± 1.8	
	Not Working	74	(14.6)	83.1 ± 14.4		9.5 ± 1.6	
Mother’s Job	Own Business	21	(4.1)	77.4 ± 20.5	0.773	9.0 ± 1.8	0.391
	Government Sector	81	(15.9)	82.0 ± 14.2		9.3 ± 1.6	
	Private Sector	10	(2.0)	76.9 ± 18.0		9.9 ± 1.6	
	Retired	19	(3.7)	80.6 ± 21.1		9.7 ± 1.6	
	HW	377	(74.2)	80.8 ± 16.8		9.6 ± 1.7	

Abbreviations: SD = standard deviation, HRQol = health-related quality of life based on the PedsQL questionnaire. *p*-value is based on the independent sample t-test or the one-way ANOVA test.

**Table 2 healthcare-11-00974-t002:** Descriptive statistics of mental health, physical activity, and quality of life for boys (n = 337) and girls (n = 264).

Scale	Component	Total		Boys		Girls		*p*-Value *	Cohen’s d
		Mean	SD	Mean	SD	Mean	SD		
DASS21 scores	Anxiety scores	4.8	5.0	3.9	4.5	5.9	5.5	<0.001	−0.414
	Depression scores	5.0	5.3	4.2	4.6	6.1	5.8	<0.001	−0.360
	Stress scores	5.5	5.5	4.7	4.9	6.4	6.1	<0.001	−0.309
FELS PAQ Total score	Leisure Index	3.1	0.8	3.1	0.9	3.1	0.7	0.922	−0.008
	Sports index	3.2	0.8	3.4	0.7	3.0	0.7	<0.001	0.553
	Work Index	3.2	1.1	3.1	1.1	3.3	1.0	0.032	−0.178
	Fles Total score	9.5	1.7	9.5	1.8	9.4	1.6	0.203	0.109
Pediatric Quality of Life (PedsQL)	Physical Functioning	83.2	17.9	86.3	17.2	79.3	18.0	<0.001	0.402
	Emotional Functioning	76.1	24.3	80.8	21.5	70.1	26.3	<0.001	0.450
	Social Functioning	85.9	19.0	87.7	17.4	83.6	20.7	0.008	0.217
	School Functioning	80.2	20.4	81.8	20.5	78.3	20.2	0.040	0.169
	Overall Qol Scores	81.4	16.9	84.1	16.0	77.8	17.5	<0.001	0.379

* *p*-value is based on the independent sample *t*-test.

**Table 3 healthcare-11-00974-t003:** The Depression, Anxiety, and Stress Scale—21 items (DASS21) (n = 601).

DASS Symptoms		N	%	HRQoL Scores		*p*-Value *	Omega Squared
				Mean	SD		
**Depression**	Normal	341	56.7	89.8	12.1	<0.001	0.400
	Mild	67	11.1	77.9	13.9		
	Moderate	99	16.5	72.0	14.2		
	Severe	45	7.5	70.7	9.1		
	Extremely Severe	49	8.2	56.3	18.9		
**Anxiety**	Normal	310	51.6	90.1	12.4	<0.001	0.348
	Mild	41	6.8	81.2	12.8		
	Moderate	105	17.5	76.5	13.4		
	Severe	49	8.2	70.9	13.5		
	Extremely Severe	96	16.0	63.9	17.9		
**Stress**	Normal	415	69.1	87.4	13.2	<0.001	0.322
	Mild	63	10.5	73.6	14.3		
	Moderate	49	8.2	72.5	14.4		
	Severe	43	7.2	62.8	15.9		
	Extremely Severe	31	5.2	56.7	18.7		

* *p*-value is based on one-way ANOVA.

**Table 4 healthcare-11-00974-t004:** Correlations between the overall quality of life, sport index, and the DASS21 domains among adolescents.

	Depression Score	Anxiety Score	Stress Score	Sport Index	Overall Quality of Life
Depression Score		0.826 **	0.849 **	−0.061	−0.643 **
Anxiety Score	0.826 **		0.839 **	−0.002	−0.603 **
Stress Score	0.849 **	0.839 **		−0.010	−0.628 **
Sport index	−0.061	−0.002	−0.010		0.062
Overall Quality of Life	−0.643 **	−0.603 **	−0.628 **	0.062	

** Correlation is significant at the 0.01 level (2-tailed).

**Table 5 healthcare-11-00974-t005:** Multiple linear regression models for the predictors of students’ quality of life.

Models		Estimate		*p*-Value	95.0% Confidence Interval for B		Model Fitness
		Beta	SE		Lower Bound	Upper Bound	
Male	(Constant)	97.61	4.40	<0.001	88.95	106.27	R^2^ = 0.424
	Depression Score	−0.53	0.15	<0.001	−0.83	−0.24	
	Anxiety Score	−0.28	0.15	0.065	−0.57	0.02	
	Stress Score	−0.30	0.14	0.036	−0.59	−0.02	
	Leisure Index	−0.84	0.95	0.379	−2.70	1.03	
	Sport index	1.11	0.98	0.260	−0.83	3.04	
	Work index	−1.71	0.71	0.017	−3.10	−0.31	
Female	(Constant)	101.98	5.30	<0.001	91.54	112.41	R^2^ = 0.462
	Depression Score	−0.63	0.15	<0.001	−0.93	−0.34	
	Anxiety Score	0.02	0.16	0.906	−0.29	0.33	
	Stress Score	−0.40	0.15	0.010	−0.70	−0.10	
	Leisure Index	−1.81	1.23	0.141	−4.23	0.61	
	Sport index	−0.03	1.14	0.979	−2.27	2.21	
	Work index	−1.77	0.91	0.052	−3.56	0.01	
All	(Constant)	99.06	3.28	<0.001	92.61	105.50	R^2^ = 0.457
	Depression Score	−0.59	0.11	<0.001	−0.80	−0.38	
	Anxiety Score	−0.15	0.11	0.168	−0.35	0.06	
	Stress Score	−0.33	0.10	0.001	−0.54	−0.13	
	Leisure Index	−1.25	0.74	0.092	−2.70	0.21	
	Sport index	0.88	0.71	0.216	−0.52	2.28	
	Work index	−1.81	0.55	0.001	−2.89	−0.72	

Abbreviation: Beta = regression coefficient; SE = standard error of the coefficient.

## Data Availability

This study has no additional supporting data to share.

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
