# Peer review of "Physical Activity, Mental Health, and Quality of Life among School Students in the Jazan Region of Saudi Arabia: A Cross-Sectional Survey When Returning to School after the COVID-19 Pandemic"

_healthcare, 2023, doi:10.3390/healthcare11070974_

Round 1

Reviewer 1 Report

Dear Authors

Many thanks for the opportunity to read this interesting manuscript. Themes of QoL and PA among adolescents are important ones. In the discussion of their manuscript the authors also an important topic about gender disparities in this context. I have few concerns about the manuscript in its current form, which the authors may wish to consider.

Firstly (and as elaborated below) conceptualizing this as a post COVID-19 research without one question being pandemic related e.g. attempt at retrospective self-evaluation if adolescents feel they have increased PA since the pandemic. In general - per definition – all research done since the pandemic can be characterized as post pandemic. Here it appears that this research provided rather an important information about PA and QoL at the given time point, but links with pandemic felt somewhat artificial as the introduction was currently constructed.

Secondly, introduction, aims, statistical methods, and discussion were not aligned. Considerable amount of new literature was introduced in the discussion with substantial emphasis in gender issues. While interesting discussion, this literature should have been included in the introduction, and the gender comparisons in aims. It would have also been good to formulate some hypotheses based on the existing literature.

Further feedback – minor suggestions are included below.

Abstract

Please could the authors consider more precision in describing all their research results. Not only that a significant relationship was found, but also the direction of the relationship. Please, could authors also consider adding how long after the COVID-19 pandemic the research was conducted. There is also a lingering concern about how the pandemic is taken in to account in this research.

Introduction

Line 45: New paragraph from PA definition?

Line 65: Does this refer to activity reductions during weekend or pandemic. Also, please could the authors add of which age groups were included in the cited research. It is known that gender and age of children/adolescent influences likelihood of e.g. physical activity.

Line 62: Reference missing

From line 67: In which context the given numbers are? Country? Please, could you consider giving a bit more context e.g. prevalence considering the numbers.

From line 71: Important information about symptoms – but how does this fit in with the your research question?

Referring to previous comment about the lingering concern of how the pandemic is taken in to account in this research. There is no e.g. comparative data from before / during pandemic. Therefore, authors should be careful how they contextualise their results in relation to pandemic. The introduction would benefit from additional literature that have considered effect of social isolation in children / adolescent to PA and QoL (if existing). This would help reader to place the results better in perspective. Looking at the questionnaires, there appeared to be no questions specific to pandemic.

Were there any hypotheses?

Methods

Please could the authors add few details about pupil specific inclusion and exclusion criteria? How was the convenience sample recruited? Was ethical clearance sought? Were pupils/parents/guardians required to provide informed consent?

Statistical methods and research aims (or hypotheses) are not well aligned. For example, statistical methods point out for group comparisons, but this is not mentioned in research aims. This should be corrected. Considering that both parametric and non-parametric analyses are used, please could the authors add whether data normality was check/needed. Finally, regarding the multiple testing, did the authors consider adjusting the level of significance?

Results

“Grades” – this can refer to either year groups or school achievement – Please consider adding clarity.

Overall, it would be good not to report e.g. percentages both in text and tables.

Line 152: Regarding e.g. the results for depression subscale. In here and methods more clarity is needed about DASS as a diagnostic tool – i.e. valid in diagnosing depression in individuals or measuring self-reported symptoms at population level. Please adjust language accordingly.  

For e.g. ANOVA tests, please could the authors e.g. line 168, report the whole test results – not just the p-value. Were post hoc comparisons completed?

Discussion

Line 219. This needs to be referenced.

From line 219: While points made in here are valid – but difficult to follow. It is not clear how authors leap from (1) traditional values to (2) lack of physical activity for girls (3) due to restrictions of traditional values to (4) pandemic restrictions and finally to (5) need for better PA provision (for girls).  This needs to be rethought.

As in elsewhere in the discussion, a lot of new information and literature as well as concepts have been introduced. These concepts and literature should have been introduced in introduction and included in research aims. This should be corrected. The discussion also appears to veer away from the actual results with repetition of points already made.

Small point is that results shouldn’t be repeated e.g. in percentages in the discussion – these are already clear from the results.  

Conclusions

Line 264: Considering the cross-sectional design – no improvements as such could be observed. However, results did not that PA was associated with higher (or better) reported QoL.

Author Response

Abstract

Please could the authors consider more precision in describing all their research results. Not only that a significant relationship was found, but also the direction of the relationship. Please, could authors also consider adding how long after the COVID-19 pandemic the research was conducted. There is also a lingering concern about how the pandemic is taken in to account in this research.

The whole abstract is modified and updated

Introduction

Line 45: New paragraph from PA definition?

Done

Line 65: Does this refer to activity reductions during weekend or pandemic. Also, please could the authors add of which age groups were included in the cited research. It is known that gender and age of children/adolescent influences likelihood of e.g. physical activity.

Modified as you suggested

Line 62: Reference missing

Modified

From line 67: In which context the given numbers are? Country? Please, could you consider giving a bit more context e.g. prevalence considering the numbers.

Done

From line 71: Important information about symptoms – but how does this fit in with the your research question?

Deleted

Referring to previous comment about the lingering concern of how the pandemic is taken in to account in this research. There is no e.g. comparative data from before / during pandemic. Therefore, authors should be careful how they contextualise their results in relation to pandemic. The introduction would benefit from additional literature that have considered effect of social isolation in children / adolescent to PA and QoL (if existing). This would help reader to place the results better in perspective. Looking at the questionnaires, there appeared to be no questions specific to pandemic.

Were there any hypotheses?

We modified the Introduction to be more directly related with the topic

Methods

Please could the authors add few details about pupil specific inclusion and exclusion criteria? How was the convenience sample recruited? Was ethical clearance sought? Were pupils/parents/guardians required to provide informed consent?

Statistical methods and research aims (or hypotheses) are not well aligned. For example, statistical methods point out for group comparisons, but this is not mentioned in research aims. This should be corrected. Considering that both parametric and non-parametric analyses are used, please could the authors add whether data normality was check/needed. Finally, regarding the multiple testing, did the authors consider adjusting the level of significance?

Thanks, actually normality test has been conducted and evaluated before conducting t and ANOVA  tests , we added a sentence for that

Results

“Grades” – this can refer to either year groups or school achievement – Please consider adding clarity.

Clarified as you suggested

Overall, it would be good not to report e.g. percentages both in text and tables.

Thank you, but we think it will add not  

Line 152: Regarding e.g. the results for depression subscale. In here and methods more clarity is needed about DASS as a diagnostic tool – i.e. valid in diagnosing depression in individuals or measuring self-reported symptoms at population level. Please adjust language accordingly.  

Adjusted  by adding symptoms,  and self-reported

For e.g. ANOVA tests, please could the authors e.g. line 168, report the whole test results – not just the p-value. Were post hoc comparisons completed?

If you mean Table 1 a new column has been added and the P value reported. If you mean Table three, just we interested weather QoL differs according to levels of mental health and it was clear that there is significant difference, between quality of life according to mental health problems, so no need for post hoc comparisons.  

Discussion

Line 219. This needs to be referenced.

Done

From line 219: While points made in here are valid – but difficult to follow. It is not clear how authors leap from (1) traditional values to (2) lack of physical activity for girls (3) due to restrictions of traditional values to (4) pandemic restrictions and finally to (5) need for better PA provision (for girls).  This needs to be rethought.

Some clarifications, were added

As in elsewhere in the discussion, a lot of new information and literature as well as concepts have been introduced. These concepts and literature should have been introduced in introduction and included in research aims. This should be corrected. The discussion also appears to veer away from the actual results with repetition of points already made.

We tried to modify the discussion to incorporate your comments

Small point is that results shouldn’t be repeated e.g. in percentages in the discussion – these are already clear from the results

We deleted some, we tried to avoid such repetition, but in some situation it is necessary  

Conclusions

Line 264: Considering the cross-sectional design – no improvements as such could be observed. However, results did not that PA was associated with higher (or better) reported QoL.

Not clear  

Reviewer 2 Report

This is a somewhat interesting paper that addresses relationships between physical activity and quality of life for school students returning to school after the Covid-19 pandemic lockdown period. In many ways, the paper presents appropriate data and seems to have been conducted diligently. Yet in certain respects the argument of the paper is not very strongly made—in particular, because the core issue stated is about the return to school, which the findings only indirectly address.

In my view, the authors should be asked to address the following issues:

·         The Abstract needs to incorporate some extra words to say (1) why the issue being studied is important and (2) what this paper adds to the literature on the topic.

·         The Introduction section needs to more directly state how the present focus (lines 74-80) is different from the previous papers (especially those discussed in lines 50-58). I guess it is the “schools” and/or “returning to schools” focus that is different here?

·         The introduction also needs to clearly and unambiguously state a research objective for the study.

·         The section on study design needs to explain why intermediate and high schools were the focus. It also needs to provide some contextual information about the Jazan region—including about how education was conducted during the lockdown period, and for how long this situation persisted.

·         Lines 97-99 contain an explanation about random and non-random sampling being used “due to the pandemic situation”. This is unclear and I hope the explanation can be rephrased and/or expanded.

·         With regard to the content on lines 103-124, I think we need to read more *justification* (rather than merely description), about the design of the instrument. What were the authors trying to do and how did their decisions on instrument design (including selecting previous instruments to use) relate to their research objectives? Were there any instrument translation issues? And were any actual changes made to the instrument as a consequence of the pilot study?

·         The Results section needs to start by signposting what will be presented, and clearly explaining how this relates to the objectives of the study. This is a fairly important point in my view—it is the apparent lack of alignment between the findings and the argument in the introduction which is my core concern as a reviewer.

·         The Discussion needs to more explicitly evaluate the strengths of the findings rather than merely summarising them. How do your findings allow you to say something new and important that the prior studies you reviewed earlier did not say? In particular, what are the implications for students upon their return to school, and/or for schools as institutions?

·         The Conclusions section would benefit from signposting (briefly) some future avenue of research suggested by the findings of this paper.

Author Response

This is a somewhat interesting paper that addresses relationships between physical activity and quality of life for school students returning to school after the Covid-19 pandemic lockdown period. In many ways, the paper presents appropriate data and seems to have been conducted diligently. Yet in certain respects the argument of the paper is not very strongly made—in particular, because the core issue stated is about the return to school, which the findings only indirectly address.

 Thanks a lot

In my view, the authors should be asked to address the following issues:

  • The Abstract needs to incorporate some extra words to say (1) why the issue being studied is important and (2) what this paper adds to the literature on the topic.

Done

  • The Introduction section needs to more directly state how the present focus (lines 74-80) is different from the previous papers (especially those discussed in lines 50-58). I guess it is the “schools” and/or “returning to schools” focus that is different here?

Modified as you suggested

  • The introduction also needs to clearly and unambiguously state a research objective for the study.

Thank you; We modified this section to make the research objectives, clearer   

  • The section on study design needs to explain why intermediate and high schools were the focus. It also needs to provide some contextual information about the Jazan region—including about how education was conducted during the lockdown period, and for how long this situation persisted.

Good point, modified as you suggested

  • Lines 97-99 contain an explanation about random and non-random sampling being used “due to the pandemic situation”. This is unclear and I hope the explanation can be rephrased and/or expanded.

We rephrased the sentence to be more understandable for journal readers

  • With regard to the content on lines 103-124, I think we need to read more *justification* (rather than merely description), about the design of the instrument. What were the authors trying to do and how did their decisions on instrument design (including selecting previous instruments to use) relate to their research objectives? Were there any instrument translation issues? And were any actual changes made to the instrument as a consequence of the pilot study?

Well, the three instruments were translated and validated in Arab population, so we did not conduct any changes. We added the internal consistency of the three measures.

  • The Results section needs to start by signposting what will be presented, and clearly explaining how this relates to the objectives of the study. This is a fairly important point in my view—it is the apparent lack of alignment between the findings and the argument in the introduction which is my core concern as a reviewer.

 The logic of presentations follows the study objectives. First it starts with describing the Physical Activity and level of quality of life according to different background characteristics, then we described the mental health status, assessing the relationship between the three variables and finally exploring the predictors.   

  • The Discussion needs to more explicitly evaluate the strengths of the findings rather than merely summarizing them. How do your findings allow you to say something new and important that the prior studies you reviewed earlier did not say? In particular, what are the implications for students upon their return to school, and/or for schools as institutions?

Modified

  • The Conclusions section would benefit from signposting (briefly) some future avenue of research suggested by the findings of this paper.

       Thanks for this point. Done

Reviewer 3 Report

This article assesses the physical activity, mental health, and quality of life of students after the COVID-19 pandemic through a sample survey and standardized questionnaire, and the data analysis methods include descriptive analysis and predictive analysis. According to the research results, the necessity of formulating intervention programs for students' physical activity levels is pointed out. However, the research methodology is not innovative enough and the research analysis is not sufficient. In order to ensure the quality of the article, extensive revisions and additions to the parts mentioned below are recommended.

1. In the introduction part of the article, the author compiled and quoted the data results from other studies for background introduction and theoretical support. However, I think it should be analyzed in depth of relevant literature and research, and add the latest research methods, so as to improve the persuasiveness of the theoretical research.

 2. At the end of the introduction part, the author should highlight the main research contributions of this work, so that readers can understand the content of the article more clearly. 

3. In the study design section, why was the student population of 12 to 18 years old selected? What was the basis for determining the age range of students?

4. In the sampling procedures section, lines 93-97, please pay attention to the format of formula writing and parameter interpretation.

5. In the data collection and instrumentation section, line 124, although the internal consistency coefficient greater than 0.7 indicates a high reliability of the questionnaire, please introduce the relevant content and results in detail so that readers can clearly understand.

6. I think the data presentation of table 1 is not complete, please add P-Value.  

7. In the third part, line 159, the paragraph beginning is lack of the phrase "Table 3".

8. In the third part, I think this is just a list of the results of questionnaires, which is not sufficiently explained and lacks a certain logic.

9. In the discussion part, the research results are compared with other studies, and perhaps the use of tables for meticulous comparative analysis will be more professional. In addition, we would like to see content on future solutions and research methods around the questionnaire results.

10. In the conclusion part, although the main contents and results of the current study are addressed, etc., they are too brief, and the importance of the research methods applied and the assessment indicators for the experimental results are not expressed. Also, we would like to see the authors' vision of the relevant intervention programs.

Author Response

This article assesses the physical activity, mental health, and quality of life of students after the COVID-19 pandemic through a sample survey and standardized questionnaire, and the data analysis methods include descriptive analysis and predictive analysis. According to the research results, the necessity of formulating intervention programs for students' physical activity levels is pointed out. However, the research methodology is not innovative enough and the research analysis is not sufficient. In order to ensure the quality of the article, extensive revisions and additions to the parts mentioned below are recommended.

  1. In the introduction part of the article, the author compiled and quoted the data results from other studies for background introduction and theoretical support. However, I think it should be analyzed in depth of relevant literature and research, and add the latest research methods, so as to improve the persuasiveness of the theoretical research.

Thank you; We modified this section to incorporate an in-depth analysis of the relationship between physical activity, mental health and quality of life.

  1. At the end of the introduction part, the author should highlight the main research contributions of this work, so that readers can understand the content of the article more clearly. 

Added as you suggested

  1. In the study design section, why was the student population of 12 to 18 years old selected? What was the basis for determining the age range of students?

Thank you, these group constitute students in the intermediate and secondary school levels. They are homogenous , we added some explanation in the methods section.

  1. In the sampling procedures section, lines 93-97, please pay attention to the format of formula writing and parameter interpretation.”

Revised as you suggested

  1. In the data collection and instrumentation section, line 124, although the internal consistency coefficient greater than 0.7 indicates a high reliability of the questionnaire, please introduce the relevant content and results in detail so that readers can clearly understand.

Ok Sir/Madam, the three instruments are validated in the Arab population, however we added out internal consistency based on Alpha for the three measures as you suggested,

. I think the data presentation of table 1 is not complete, please add P-Value.  

Thanks, Good suggestion, We added new two columns and added the p value

  1. In the third part, line 159, the paragraph beginning is lack of the phrase "Table 3".

Modified

  1. In the third part, I think this is just a list of the results of questionnaires, which is not sufficiently explained and lacks a certain logic.

OK, no Sir/Madam, the logic of presentations follows the study objectives. First it starts with describing the Physical Activity and level of quality of life according to different background characteristics, then we described the mental health status, assessing the relationship between the three variables and finally exploring the predictors.   

  1. In the discussion part, the research results are compared with other studies, and perhaps the use of tables for meticulous comparative analysis will be more professional. In addition, we would like to see content on future solutions and research methods around the questionnaire results.

Thanks, using Tables in the discussion section will increase the number of Tables in the manuscript and may affect the readers, so excuse us, Sir/Madam not adding them; however, the discussion has been updated and revised. We added future solutions and research methods around the questionnaire results. As you suggested

  1. In the conclusion part, although the main contents and results of the current study are addressed, etc., they are too brief, and the importance of the research methods applied and the assessment indicators for the experimental results are not expressed. Also, we would like to see the authors' vision of the relevant intervention programs.

Thanks a lot for this point; we modified the conclusion section according to your suggestions.

Reviewer 4 Report

I would like to thank the authors and the Editorial Board for the opportunity to review the article submitted to Healthcare. The authors' manuscript refers to a very important topic: the quality of life and mental health during the COVID-19 pandemic. The authors’ manuscript presents very interesting results on a Saudi Arabian sample. I believe that some minor changes will improve the overall quality of their manuscript.

Introduction: The presented introduction is very short, and doesn’t provide sufficient rationale for the study. The authors do not refer to any known theory/model about quality of life or mental health,  pandemic stress and anxiety. I highly recommend that authors extend that section based on any known theoretical model and refer to more experimental and longitudinal research studies – as it stands, their manuscripts’ introduction is mostly based on cross-sectional data.

Results: Authors should calculate and report effect size measures. P-values are highly correlated with the size of the sample (see Lakens, 2022: Sample size justification). The authors present results on a fairly large sample size, therefore most of the tested differences would be significant by default. Therefore,  I highly recommend that the authors report effect size measures (which are less biased than p-values). For chi-squared analysis, authors can report Cramer’s V (for 2xX tables) or Yule’s Phi (for 2x2 tables), for t-test Cohen’s d can be calculated, and for ANOVA authors can present the results of omega-squared. See Tomczak & Tomczak (2014).

Results: Authors do not report many important test statistics. Test statistics are a significant source of information, which allows the reader or the reviewer to verify if the authors’ manuscript is free of any p-hacking manipulations. Please, report all of the mentioned statistics for the used tests:

·         For ANOVA: F-value, df1 and df2, omega-squared

·         For t-test: t-value, df, Cohen’s d

·         For chi-squared: chi-squared, df, observed N and expected N, Cramer’s V or Yule’s Phi.

Results: Table 5 presents the results of the regression analysis. I highly recommend that the authors perform an invariance analysis, comparing the tested models between two tested groups. Without proper measurement invariance verification, it is impossible to assess if the tested models are comparable or significantly different.

Discussion: I highly recommend that the authors re-write their discussion based on the calculated effect size measures.

Author Response

I would like to thank the authors and the Editorial Board for the opportunity to review the article submitted to Healthcare. The authors' manuscript refers to a very important topic: the quality of life and mental health during the COVID-19 pandemic. The authors’ manuscript presents very interesting results on a Saudi Arabian sample. I believe that some minor changes will improve the overall quality of their manuscript.

Thank you, Sir, for these comments

Introduction: The presented introduction is very short, and doesn’t provide sufficient rationale for the study. The authors do not refer to any known theory/model about quality of life or mental health,  pandemic stress and anxiety. I highly recommend that authors extend that section based on any known theoretical model and refer to more experimental and longitudinal research studies – as it stands, their manuscripts’ introduction is mostly based on cross-sectional data.

  Thank you;  We modified this section to incorporate an in-depth analysis of the relationship between physical activity, mental health and quality of life.

Results: Authors should calculate and report effect size measures. P-values are highly correlated with the size of the sample (see Lakens, 2022: Sample size justification). The authors present results on a fairly large sample size, therefore most of the tested differences would be significant by default. Therefore, I highly recommend that the authors report effect size measures (which are less biased than p-values). For chi-squared analysis, authors can report Cramer’s V (for 2xX tables) or Yule’s Phi (for 2x2 tables), for t-test Cohen’s d can be calculated, and for ANOVA authors can present the results of omega-squared. See Tomczak & Tomczak (2014).

Thank you conducted as you suggested

Results: Authors do not report many important test statistics. Test statistics are a significant source of information, which allows the reader or the reviewer to verify if the authors’ manuscript is free of any p-hacking manipulations. Please, report all of the mentioned statistics for the used tests:

  • For ANOVA:F-value, df1 and df2, omega-squared
  • For t-test: t-value, df, Cohen’s d
  • For chi-squared:chi-squared, df, observed N and expected N, Cramer’s V or Yule’s Phi.

Cohen’s d and omega-squared were Calculated as you suggested by adding a new column to Table 2 and Table 3. We did not use Chi-square. In Table one, only a few relationships are significant; add to that it was not possible to add other columns to the Table. 

Results: Table 5 presents the results of the regression analysis. I highly recommend that the authors perform an invariance analysis, comparing the tested models between two tested groups. Without proper measurement invariance verification, it is impossible to assess if the tested models are comparable or significantly different.

Thank you Sir/Madam we were not able to conduct further analysis regarding the invariance analysis, comparing the tested models between two tested groups. We did not want to flood the manuscript with mathematics, while the method might not be meaningful and valid comparisons across groups [1,2].  So, we apologize for this point. 

Discussion: I highly recommend that the authors re-write their discussion based on the calculated effect size measures.

We calculated the effects size in Table 2 and Table 3, and we found that effect size measures support the significant results, however, the whole section has been updated

1.Robitzsch A, Lüdtke O. Why Measurement Invariance is Not Necessary for Valid Group Comparisons.

  1. Welzel, C., Brunkert, L., Kruse, S., & Inglehart, R. F. (2021). Non-invariance? An Overstated Problem With Misconceived Causes. Sociological Methods & Research0(0).

Round 2

Reviewer 1 Report

Many thanks for the opportunity to review the re-drafted manuscript. I would also like to apologisise to the authors that in my original feedback the last comments was not clear - it was no wonder as a typo meant that instead of "note" - I wrote "not".

The manuscript has considerably improved and I would have only two small comments for authors to consider.

1. Study objectives would benefits from further sharpening, especially in relation to what is meant by the mechanisms underlying the associations. It appears that interactions with gender are considered as a part of the mechanisms.

2. from line 251 "moreover, decline..." This in unclear. Does this refer to results from this research or other previous studies? Authors should consider redrafting the start of the paragraph, as this gives the impression that  a directional hypothesis with a longitudinal data was tested.  Furthermore, authors contrast the previous study with gender differences found in here, making the chain of argumentation unclear.

Author Response

Many thanks for the opportunity to review the re-drafted manuscript. I would also like to apologisise to the authors that in my original feedback the last comments was not clear - it was no wonder as a typo meant that instead of "note" - I wrote "not".

Thanks a lot

The manuscript has considerably improved and I would have only two small comments for authors to consider.

Thanks 

  1. Study objectives would benefit from further sharpening, especially in relation to what is meant by the mechanisms underlying the associations. It appears that interactions with gender are considered as a part of the mechanisms.

 The research objectives were further modified to relate to the overall outcome directly.  

  1. from line 251 "moreover, decline..." This in unclear. Does this refer to results from this research or other previous studies? Authors should consider redrafting the start of the paragraph, as this gives the impression that  a directional hypothesis with a longitudinal data was tested.  Furthermore, authors contrast the previous study with gender differences found in here, making the chain of argumentation unclear.

Thank you, Sir, for this point; we revised the paragraph and added a new reference supporting the point. Now no relation between the paragraph and the gender comparisons. 

Reviewer 4 Report

I find authors answers to the review report satisfactory

Author Response

Thanks a lot